In vitro culture of leukemic cells in collagen scaffolds and carboxymethyl cellulose-polyethylene glycol gel

http://orcid.org/0000-0002-0025-4364 Svozilova Hana 1 2 3
Vojtova Lucy 4
Matulova Jana 4
Bruknerova Jana 1 3
http://orcid.org/0009-0001-0919-5313 Polakova Veronika 4
http://orcid.org/0000-0001-8103-148X Radova Lenka 1 3
http://orcid.org/0000-0002-1269-6282 Doubek Michael 1 2 3
http://orcid.org/0000-0002-6148-8877 Plevova Karla 1 2 3 karla.plevova@mail.muni.cz
http://orcid.org/0000-0001-7136-2680 Pospisilova Sarka 1 2 3 sarka.pospisilova@ceitec.muni.cz
1 Center of Molecular Medicine, Central European Institute of Technology, Masaryk University , Brno , Czech Republic
2 Department of Internal Medicine-Hematology and Oncology, University Hospital Brno and Faculty of Medicine, Masaryk University , Brno , Czech Republic
3 Institute of Medical Genetics and Genomics, University Hospital Brno and Faculty of Medicine, Masaryk University , Brno , Czech Republic
4 Advanced Biomaterials, Central European Institute of Technology, Brno University of Technology , Brno , Czech Republic
Lonardo Enza
Electronic publication date: 2024 Dec 6
Publication date: 2024
Volume: 12
Electronic Location ID: e18637
Received 2024 Jun 17; Accepted 2024 Nov 13
Copyright: © 2024 Svozilova et al.
Copyright year: 2024
Copyright holder: Svozilova et al.
License: This is an open access article distributed under the terms of the Creative Commons Attribution License, which permits unrestricted use, distribution, reproduction and adaptation in any medium and for any purpose provided that it is properly attributed. For attribution, the original author(s), title, publication source (PeerJ) and either DOI or URL of the article must be cited.
License URL: https://creativecommons.org/licenses/by/4.0/

Keywords: Chronic lymphocytic leukemia, CLL, 3D culture, Carboxymethyl cellulose, CMC, Polyethylene glycol, PEG, Collagen, Scaffolds, Gel

Funding: Ministry of Health, Czech Republic DRO FNBr65269705 National Institute for Cancer Research LX22NPO5102 European Union - Next Generation EU Johannes Amos Comenius Programme called Excellent Research CZ.02.01.01/00/22_008/0004562 MEYS CR EATRIS-CZ LM2023053 NCMG LM2023067 Czech-BioImaging LM2023050 CzechNanoLab LM2023051 This work was supported by projects DRO FNBr65269705 (Ministry of Health, Czech Republic), the National Institute for Cancer Research LX22NPO5102 (Programme EXCELES, funded by the European Union - Next Generation EU), and the project EXRegMed no. CZ.02.01.01/00/22_008/0004562 funded by Johannes Amos Comenius Programme called Excellent Research. Research data were generated in collaboration with the CEITEC core facilities Genomics, Cellular Imaging, and Nano, operating within infrastructures funded by MEYS CR, namely EATRIS-CZ (LM2023053), NCMG (LM2023067), Czech-BioImaging (LM2023050), and CzechNanoLab (LM2023051). There was no additional external funding received for this study. The funders had no role in study design, data collection and analysis, decision to publish, or preparation of the manuscript.

==============================
Background

Chronic lymphocytic leukemia (CLL) is a common adult leukemia characterized by the accumulation of neoplastic mature B cells in blood, bone marrow, lymph nodes, and spleen. The disease biology remains unresolved in many aspects, including the processes underlying the disease progression and relapses. However, studying CLL in vitro poses a considerable challenge due to its complexity and dependency on the microenvironment. Several approaches are utilized to overcome this issue, such as co-culture of CLL cells with other cell types, supplementing culture media with growth factors, or setting up a three-dimensional (3D) culture. Previous studies have shown that 3D cultures, compared to conventional ones, can lead to enhanced cell survival and altered gene expression. 3D cultures can also give valuable information while testing treatment response in vitro since they mimic the cell spatial organization more accurately than conventional culture.

Methods

In our study, we investigated the behavior of CLL cells in two types of material: (i) solid porous collagen scaffolds and (ii) gel composed of carboxymethyl cellulose and polyethylene glycol (CMC-PEG). We studied CLL cells’ distribution, morphology, and viability in these materials by a transmitted-light and confocal microscopy. We also measured the metabolic activity of cultured cells. Additionally, the expression levels of MYC, VCAM1, MCL1, CXCR4, and CCL4 genes in CLL cells were studied by qPCR to observe whether our novel culture approaches lead to increased adhesion, lower apoptotic rates, or activation of cell signaling in relation to the enhanced contact with co-cultured cells.

Results

Both materials were biocompatible, translucent, and permeable, as assessed by metabolic assays, cell staining, and microscopy. While collagen scaffolds featured easy manipulation, washability, transferability, and biodegradability, CMC-PEG was advantageous for its easy preparation process and low variability in the number of accommodated cells. Both materials promoted cell-to-cell and cell-to-matrix interactions due to the scaffold structure and generation of cell aggregates. The metabolic activity of CLL cells cultured in CMC-PEG gel was similar to or higher than in conventional culture. Compared to the conventional culture, there was (i) a lower expression of VCAM1 in both materials, (ii) a higher expression of CCL4 in collagen scaffolds, and (iii) a lower expression of CXCR4 and MCL1 (transcript variant 2) in collagen scaffolds, while it was higher in a CMC-PEG gel. Hence, culture in the material can suppress the expression of a pro-apoptotic gene (MCL1 in collagen scaffolds) or replicate certain gene expression patterns attributed to CLL cells in lymphoid organs (low CXCR4, high CCL4 in collagen scaffolds) or blood (high CXCR4 in CMC-PEG).

Introduction

Chronic lymphocytic leukemia (CLL) is the most common adult leukemia in Caucasian populations. The average yearly incidence in Western countries is 4.2/100,000 inhabitants and reaches more than 30/100,000 in people over 80 years of age (Eichhorst et al., 2021). Over the last five decades, the 5-year survival of CLL patients has increased by 22 percentage points with an estimate of 87.2% (Hallek & Al-Sawaf, 2021). While CLL is generally a slow-growing cancer, it can develop treatment-resistant and life-threatening complications. Nevertheless, advances in CLL diagnosis and management have improved outcomes for many patients, and ongoing research aims to develop new, more effective treatments.

CLL is a disease of neoplastic mature B cells accumulating in blood, bone marrow, lymph nodes, and spleen (Hallek & Al-Sawaf, 2021). However, the laboratory research of CLL biology has been challenging due to the complexity of the disease and the dependency of CLL cells on their microenvironment. When CLL cells are cultured without supplementing their natural interactions, they undergo apoptosis shortly after their removal from a patient’s body (Burgess et al., 2012; Hoferkova, Kadakova & Mraz, 2022). Few CLL immortalized cell lines have been established to overcome this issue, e.g., MEC-1, HG-3, PGA-1, OSU-CLL, or PCL12. Although the existing CLL cell lines cover a large number of genetic aberrations, they still do not accurately reflect the full heterogeneity of the disease. So far, no commercially available cell lines bear defects, such as deletion 11q or mutations in ATM, SF3B1, or NOTCH1 genes, despite their frequent occurrence in CLL patients. Moreover, their immortalization might have led to dysregulation of critical biological pathways involved in CLL pathogenesis; thus, their behavior might deviate from reality (Lanemo Myhrinder et al., 2013; Agathangelidis et al., 2015).

Therefore, numerous solutions were proposed to mimic the original niche of primary B cells (Crassini et al., 2017; Scielzo & Ghia, 2020). CLL cells can be co-cultured with other naturally co-occurring cell types, such as bone marrow stromal cells (BMSCs), nurse-like cells, or T cells (Panayiotidis et al., 1996; Kurtova et al., 2009). The culture medium can also be supplemented with peptides and growth factors, such as interleukins or CD40 ligand (CD40L), beneficially imitating the presence of cells (Panayiotidis et al., 1993; Kitada et al., 1999; Pascutti et al., 2013; Herman & Wiestner, 2016). Another advancement is to culture the cells in a three-dimensional (3D) setting since conventional (so-called two-dimensional or 2D) cultures often fail to replicate the cellular and extracellular interactions occurring in vivo, leading to discrepancies between in vitro and in vivo results. 3D cultures can better mimic the in vivo microenvironment, allowing for more accurate evaluation of drug efficacy and toxicity.

Various 3D in vitro models of hematological malignancies have been developed, e.g., for acute myeloid leukemia (Blanco et al., 2010; Aljitawi et al., 2014; Shen et al., 2016; Bray et al., 2017; Karimpoor et al., 2018), multiple myeloma (Ferrarini et al., 2013; Belloni et al., 2018, 2022; Wu et al., 2022), acute lymphoblastic leukemia (Bruce et al., 2015), or chronic lymphocytic leukemia (Barbaglio et al., 2020; Sbrana et al., 2021; Svozilová et al., 2021; Belloni et al., 2022; Haselager et al., 2020, 2023; Ribezzi et al., 2023). Some of these studies have shown that 3D cultures can alter gene expression (Sbrana et al., 2021) or response to treatment (Aljitawi et al., 2014; Shen et al., 2016; Wu et al., 2022). Moreover, compared to conventional culture, 3D-cultured neoplastic lymphoid or myeloid cells can exhibit longer survival rates (Haselager et al., 2020, 2023; Sbrana et al., 2021; Wu et al., 2022; Ribezzi et al., 2023).

In our research, we have utilized collagen scaffolds and a gel mixture of carboxymethyl cellulose (CMC) with polyethylene glycol (PEG; combination abbreviated as CMC-PEG). Collagen is a major component of the extracellular matrix (ECM) and is known to play a crucial role in the development and progression of cancer (Xiong & Xu, 2016). CMC is a water-soluble cellulose derivative commonly used as a thickener, stabilizer, and emulsifier in various industrial and food applications. In biomedical research, CMC has also been introduced into cell culture (Aravamudhan et al., 2014; Tavakol et al., 2021). Polyethylene glycol (PEG) is a synthetic, water-soluble polymer widely utilized in various biomedical applications, including drug delivery and tissue engineering (Kong et al., 2017).

Our study aimed to explore the behavior of bone marrow stromal and CLL cell lines and genetically characterized primary CLL cells in the abovementioned materials. Specifically, we focused on cell viability, metabolic levels, morphology, and changes in gene expression in 3D culture compared to conventional culture. We hypothesized that the culture in our materials could increase the viability and metabolic levels of primary CLL cells, as well as the expression of anti-apoptotic genes and genes associated with cell-to-cell interactions. Additionally, we aimed to discover practical aspects of working with the materials and provide recommendations for methodology in future studies.

Materials and Methods

Preparation of collagen scaffolds

Freeze-dried bovine collagen (Collado, Brno, Czech Republic) was diluted from 100% to 0.5% (w/v) in ultrapure water and left swelling at 4 °C for 2 h. The mixture was disintegrated with a disperser (T 18 digital ULTRA-TURRAX; IKA, Staufen im Breisgau, Germany) for 5–10 min at 6,500 rpm at 4 °C until homogenized. Subsequently, 150 μL of the swelled disintegrated collagen was pipetted into each well of a 96-well plate (Techno Plastic Products, Trasadingen, Switzerland). The 0.5% (w/v) collagen was freeze-dried in a lyophilizer (EPSILON 2–10; Martin Christ, Osterode am Harz, Germany) at −35 °C under 1 mBar for 15 h followed by a secondary drying process at 25 °C under 0.01 mBar until decreasing Δp was up to 10%. Afterwards, the scaffolds were cross-linked either for 5 or 120 min with 150 μL solution of 25 nM N-(-3-Dimethylaminopropyl)-N’-ethyl carbodiimide hydrochloride (EDC) and 12.5 nM N-hydroxysuccinimide (NHS) diluted in 96% ethanol. The two incubation times were initially tested to choose the optimum for subsequent experiments. The cross-linking reaction was terminated by adding 2 × 200 μL of 0.1 M disodium phosphate (Na2HPO4) per well. The scaffolds were washed three times with 200 μL of ultrapure water, each time for 20–30 min. Then, stabilized materials were freeze-dried once again. Finally, they were sterilized by ethylene oxide at 37 °C for 5 h. The ethylene oxide residues were vented from the scaffolds at least for 24 h, and the resulting scaffolds were stored at room temperature.

Preparation of CMC-PEG gel

The gel was prepared by mixing 6% (w/v) CMC with 5% (w/v) PEG 1,500 in ultrapure water. The mixture was incubated at 120–140 °C for 30 min. Prepared CMC-PEG gel was stored at −20 °C. Prior to use, it was weighed and subsequently sterilized in an open dish by ultraviolet light in a laminar flow cabinet for 30 min. Then, the gel was transferred into a sterile 50 mL falcon tube, and medium was added in a ratio 1:3 (gel:medium). The mixture was stirred until completely homogeneous. Before evaluating the homogeneity of the mixture, the tubes with gel were briefly centrifuged. Finally, the 1:3 CMC-PEG:medium mixture was added to cells in various ratios. Specifically, the stock gel was diluted 60, 24, 12, or 8 times, creating 1.67%, 4.17%, 8.3%, and 16.6% gel compared to the stock solution. Since the stock gel contains 60 g/L (w/v) CMC and 50 g/L (w/v) PEG, the resulting concentrations after dilutions were 1.0 g/L; 2.5 g/L, 5.0 g/L or 7.5 g/L (w/v) for CMC, and 0.83 g/L; 2.08 g/L; 4.17 g/L; or 6.25 g/L (w/v) for PEG, respectively. The mixture was thoroughly homogenized by pipetting with a 1 mL wide bore tip. Since CMC-PEG was previously dissolved in water, all controls without gel always contained a corresponding amount of water.

Cell lines and primary CLL cells

Commercially available cell lines, as well as primary patient CLL cells, were used. The cell lines included adherent bone marrow stromal cell lines of human or murine origin, namely HS-5 (CRL-11882; ATCC, Manassas, VA, USA) or M2-10B4 (CRL-1972; ATCC). As for CLL cell lines, MEC-1 (DSMZ ACC 497; Braunschweig, Germany) or HG-3 (Rosén et al., 2012) (kindly provided by Prof. Richard Rosenquist) were used. Vitally frozen primary CLL cells were selected from the institutional biobank, which stores samples of patients monitored at the Department of Internal Medicine, Hematology and Oncology, University Hospital Brno, Czech Republic. All included patients provided written informed consent with the use of their samples for research purposes. The project was approved by the institutional ethical committee under the registration number SUp 8/18. Neoplastic B cells were separated from peripheral blood with high purity (>98%) using Ficoll-Paque Plus (GE Healthcare, Uppsala, Sweden) coupled with RosetteSep human B cell enrichment cocktail and CD3+ cell depletion cocktail (StemCell Technologies, Vancouver, Canada). The purity of the cells was assessed by flow cytometry. Clinical and laboratory parameters were known for each sample, including genetic aberrations analyzed either by fluorescence in situ hybridization (FISH) or by sequencing, i.e., mutational statuses of the variable region of the immunoglobulin heavy chain gene (IGHV), TP53, and NOTCH1 genes (Table S1).

All primary cells and cell lines were maintained in culture at 37 °C in the 5% CO2 atmosphere in media recommended by the cells resource center (Table S2). Each medium was supplemented with a heat-inactivated (56 °C for 30 min) 10% South American fetal bovine serum (FBS; catalog number FB-1001B/500), 100 I.U./mL penicillin and 100 μg/mL (w/v) streptomycin (Biosera, Nuaille, France). To mimic the interaction with T cells (Hoferkova, Kadakova & Mraz, 2022), interleukin 4 (IL-4) and CD40L (both Thermo Fisher Scientific, Carlsbad, CA, USA) were optionally added to the medium, reaching final concentrations of 20 ng/mL (w/v) and 100 ng/mL (w/v), respectively (Lezina et al., 2018). Cell lines were passaged two or three times a week after reaching approximately 80% confluence. Additionally, they were routinely tested by MycoAlert Plus Detection Kit (Lonza, Basel, Switzerland) to exclude Mycoplasma infection.

Establishment and maintenance of the 3D (co-)culture

The number of cells used for the experiments was determined using Luna-FL automated fluorescence cell counter (Logos Biosystems, Anyang, South Korea). Prior to counting, cells were stained with acridine orange (membrane permeable; staining all nuclei) and propidium iodide (membrane impermeant; marking only nuclei of dead cells). A population of viable cells was enriched by the EasySep Dead Cell Removal (Annexin V) Kit (StemCell Technologies, Vancouver, Canada) according to the manufacturer’s protocol. After the enrichment, only the number of viable cells was considered for the further experiment setup. Different cell types were seeded in different amounts per well (with material or without material as a 2D control), considering material properties. The seeding densities were the following: adherent cell lines at 5 × 104 per well, suspension cell lines at 1 or 2 × 105 per well, primary CLL cells at 8 × 105 or 1.6 × 106 cells per well. Twofold higher seeding densities were applied when non-adherent cells were seeded into collagen scaffolds to compensate for the cells flowing outside the scaffold.

The CMC-PEG gel was mixed with the cells by pipetting. The collagen scaffolds were seeded with cells according to the previously published method (Passaro et al., 2017) with slight modifications. The freeze-dried scaffolds were transferred to 96-well plate by needle. Then, 10 μL of cell suspension was pipetted into each scaffold, making it fully reconstituted and seeded. When working with adherent cells, the plate with scaffolds was placed into an incubator (37 °C, 5% CO2) for 1 h to let the adherent cells attach to the scaffold; afterward, the medium was added to each well to cover the whole scaffold. When cells growing in suspension were cultured in the scaffold, the medium fully covering the scaffold was added immediately. When the cell lines were cultured for a long term (minimum of 3 days), whole populated scaffolds were transferred into a fresh medium every 3 to 4 days without trypsinization, cell dilution, or washing.

While co-culturing M2-10B4 and primary CLL cells, the M2-10B4 cells were seeded into the scaffolds/gel 24 h before the primary CLL cells. According to the previously published protocol (Passaro et al., 2017), the excess liquid from collagen scaffolds with M2-10B4 cells was first removed; then, primary CLL cells were added. Afterward, the co-culture continued for 48 h until the cells were monitored by confocal microscopy or harvested for RNA isolation.

AlamarBlue assay

The metabolic activity of cells was determined by the AlamarBlue viability assay (Thermo Fisher Scientific, Eugene, OR, USA) following the manufacturer’s protocol. Solid porous collagen scaffolds were transferred to new wells containing 100 μL of diluted reagent to exclude cells growing outside the scaffolds from measurement. For cells cultured in CMC-PEG gel or cells grown in suspension, the reagent (1/10 of total volume) was added directly to the same wells where the cells were cultured. The incubation continued for 4 h at 37 °C in a cell culture incubator protected from direct light. Then, the fluorescence of metabolic products (relative fluorescence units, RFU) was measured by the microplate reader Spark 10 M (Tecan, Männedorf, Switzerland) with the excitation and emission wavelengths of 460 nm and 590 nm, respectively.

Collagen scaffolds were removed from the wells, and the solution alone was measured. As AlamarBlue should be nontoxic, continuous monitoring of the same scaffold and its repetitive exposure to AlamarBlue was possible. Thus, the scaffolds removed from the AlamarBlue solution were washed with PBS and placed into the fresh complete medium to continue the culture. The cell-free medium and unseeded materials were treated the same way as 3D- or 2D-cultured cells and served as negative controls.

Microscopy

The conventional culture of cells, unseeded scaffolds, and cells in CMC-PEG gel were checked using the EVOS FL microscope (AMG, Bothell, WA, USA; only transmission channel) or by confocal microscope Zeiss LSM 800 (Zeiss Group, Oberkochen, Germany; live-dead staining).

Two types of live-dead staining were used for the monitoring of cell viability: In monoculture, cells were stained by calcein acetoxymethyl (AM) (1:1,000), propidium iodide (1:1,000) and Hoechst 33342 (1:2,000). These were diluted in a medium without FBS and incubated at 37 °C for 45 min under the 5% CO2 atmosphere. Afterward, scaffolds were placed into the eight-well chambered cover glass system (catalog number C8-1.5H-N; Cellvis, Mountain View, Canada) into 50 μL of PBS.

In the co-culture of two cell types, a population of M2-10B4 was labeled with the CellTrace CFSE Cell Proliferation kit, and primary CLL cells were stained with CellTrace Violet before adding them to M2-10B4 cells. The co-cultures were set up in the eight-well imaging chambers already from the beginning of the experiment (i.e., seeding of M2-10B4 cells). Dead cells were marked by propidium iodide (1:1,000) straight before imaging (5 min maximum) to avoid its toxic effect on cells. All dyes were obtained from Thermo Fisher Scientific (Waltham, MA, USA).

Image processing and analysis

Three-dimensional images acquired from the confocal microscope were converted to one layer in Zeiss ZEN Lite software (version 2.3, blue edition). The conversion was performed by the Orthogonal Projection method with the frontal projection plane (XY), choosing the maximum signal intensity from all layers for each pixel. The viability of cells was measured in ImageJ (version 1.54c) (Schindelin et al., 2012). Monocultures were analyzed according to the previously published pipeline (Svozilová et al., 2021). When primary CLL cells were co-cultured with the M2-10B4 cell line, the viability of CLL cells was estimated through the following steps (for exact commands, see Table S3): Channels (channel 1 for all CLL cells, channel 2 for dead cells of both co-cultured cell types) were converted to binary image via the Otsu’s method for automatic thresholding. Then, the selection of foreground (i.e., all cells in the channel) was created and saved as a region of interest (ROI) for each channel. An overlap of the two ROIs from both channels was performed, and an area of each ROI was measured. Then, the area of overlap’s ROI was divided by the area of CLL cells’ ROI; the result represented the percentage of dead cells.

RNA isolation

Collagen scaffolds were transferred by tweezers into a microtube with 500 μL of TRI Reagent (Molecular Research Center, Cincinnati, OH, USA) and thoroughly mixed to ensure cell lysis in all layers of the scaffold. Cells cultured in CMC-PEG gel were firstly transferred to a 1.5 mL tube (whole volume, i.e., 200 μL), and the culture plate well was washed with fresh 4 × 250 μL of PBS, which was added to the 1.5 mL tube with CMC-PEG gel to maximize the input and to further dilute the gel. Then, the tubes were centrifuged at 4 °C at 390 g for 15 min, 1 mL of supernatant was discarded, and the cells were lysed with 500 μL of TRI Reagent. Cells lysed in TRI Reagent were stored at −80 °C for up to 1 month until further procedure. RNA isolation was then performed according to the manufacturer’s protocol with the following variables: 2 μL of glycogen was added as a precipitation carrier; RNA was eluted in 40 μL of nuclease-free water. Concentration was measured using NanoDrop 2000c (Thermo Scientific, Waltham, MA, USA).

qPCR

Primers amplifying the following genes were used: HPRT1 and GUSB as endogenous controls, MYC, VCAM1, transcript variant 2 of MCL1, CXCR4, and CCL4. Human-specific primers were designed for the genes to analyze their expression specifically in patients’ primary CLL cells when co-cultured with murine M2-10B4 stromal cells. Further details about primer design and evaluation are summarized in Supplementary Methods (Fig. S1, Table S4, File S1).

The qPCR was performed with the Luna universal one-step RT-qPCR kit (New England Biolabs, Ipswich, MA, USA). Reactions (10 μL) were prepared according to the reaction setup recommended in the manufacturer’s protocol and carried out using the QuantStudio 12K Flex Real-Time PCR System (Applied Biosystems, Waltham, MA, USA). Data were analyzed in the QuantStudio 12K Flex Software, v1.4.

Statistical evaluation

Graphs were created using GraphPad Prism software, version 9.5.1 for Windows (San Diego, CA, USA, www.graphpad.com). All plotted values represent the medians of specific measurements and their ranges. Gene expression data were evaluated as follows: The linear models with predictors type of culture, IGHV mutational status, TP53, and NOTCH1 mutation status were applied for each gene (MYC, VCAM1, MCL1, CXCR4, CCL4), P-values were further estimated by the empirical Bayes approach. P-values below 0.05 were considered statistically significant.

Results

Material characterization

Dry collagen scaffolds were of a white color and spongy structure. As they were prepared in the 96-well plate, they formed cylinders with an approximate diameter of 4 mm and a height of 2 mm (Fig. 1A). The scaffolds were made up of microporous and fibrous sectors ranging from 50 to 200 μm in size (Fig. 1B). Their integrity allowed for easy handling and transferability by tweezers or a needle without causing damage to the scaffold structure. Upon reconstitution in the medium, the scaffolds became translucent, enabling their observation through transmitted light and confocal microscopy (Figs. 1C–1E). Advantageously, the scaffolds did not manifest non-specific staining by the dyes used in this study.

Figure 1 Images of unpopulated materials.

(A, B) lyophilized and (C–E) reconstituted collagen scaffolds, (F) 1 g of stock CMC-PEG gel, (G) Medium without and (H) with 24× diluted CMC-PEG gel. Scaffolds were reconstituted in complete RPMI-1640 medium. Captured by (A, C, F–H) Samsung SM-A13F/DSN, (B) scanning electron microscope Tescan Mira 3, (D, E) EVOS FL microscope, transmission channel.

CMC-PEG gel formed a transparent viscous substance (Fig. 1F). Working with 7.5 g/L (w/v) CMC and 6.25 g/L (w/v) PEG and higher concentrations of CMC-PEG gel required pipetting with wide-bore pipette tips, whereas lower concentrations could be treated with regular pipette tips. The gel was permeable for all solutions used. Gel added to the medium did not alter the medium color or transparency (Figs. 1G, 1H). Similar to collagen scaffolds, CMC-PEG did not manifest non-specific staining by any dyes. Material characterization and practical aspects of working with both materials are evaluated in Table 1.

Table 1 Comparison of collagen scaffolds and CMC-PEG gels.

Material	Origin	Preparation	Sterilization	Manipulation	Washability	Variability	
Collagen scaffolds	Natural (bovine collagen)	Disintegration → freeze-drying → cross-linking → freeze-drying	Ethylene oxide or gamma irradiation	Tweezers	Continuous experiments possible–scaffolds are washable and transferrable	Variable–each material has slightly different pore distribution and orientation;
an unknown number of cells successfully seeded in the scaffolds (cell losses due to cells partly flowing outside the scaffold)	
CMC-PEG gel	CMC–natural;
PEG–synthetic	Powder + H2O → 120-140 °C for 30 min	UV light	Weighing, wide-bore tip pipetting	Difficult to wash (all additives mostly remain in the gel)	Each material displays the same features;
a known number of cells seeded into the gel	
Note:

Washability refers to the ability of materials to be washed after exposure to a substance, which is non-toxic for a short period of time but can be toxic after long-exposure (e.g., AlamarBlue).

Culture in collagen scaffolds

The pore size in collagen scaffolds was sufficient for accommodating bone marrow stromal cells (~15–20 μm) as well as leukemic B cells (~8–12 μm)–this was initially confirmed by the 3D culture of corresponding cell lines. Their behavior in collagen scaffolds was studied either by the AlamarBlue assay (Fig. 2) or confocal microscopy (Figs. 3E–3H). HS-5, M2-10B4, MEC-1, and HG-3 cell lines were able to maintain stable levels of metabolism for a minimum of 60 days when cultured in the 120-min-crosslinked collagen scaffolds (Fig. 2). The cells in the scaffolds did not have to be sub-cultured (e.g., trypsinized or diluted) and supplying them with the fresh complete medium was sufficient to maintain their metabolic levels. If 5 min of cross-linking was used during the material preparation, it led to scaffolds that dissolved after 25 days in the presence of HS-5 cells. Thus, only scaffolds formed after cross-linking lasting 120 min were used for further experiments, as they preserved the structure in the presence of all cell lines for a minimum of 2 months. This observation proved that the scaffolds were biodegradable but at the same time, their structure could be intact for sufficient time during in vitro experiments.

Figure 2 Metabolic levels of HS-5, M2-10B4, MEC-1, and HG-3 cell lines growing in collagen scaffolds.

Scaffolds were prepared with cross-linking for 120 (black) or 5 min (magenta). Measured by AlamarBlue, RFU, relative fluorescence units. N = 3 biological replicates.

Figure 3 Micrographs of cell lines in different types of culture.

Cells cultured (A–D) conventionally, (E–H) in collagen scaffolds or (I-L) 24× diluted CMC-PEG gel. Captured 3 days after seeding by (A–D, I–L) EVOS FL microscope, transmission channel, (E–H) confocal microscope Zeiss LSM 800, green–live cells (calcein AM), red–dead cells (propidium iodide), blue–all nuclei (Hoechst 33342).

It was also observed that cells growing in suspension were partly flowing outside the collagen scaffolds, both during their culture and each scaffold transfer to a new well, causing a remarkable cell loss. In contrast to the immortalized cell lines, primary CLL cells did not compensate for such a decrease in proliferation. Hence, we show no results of the AlamarBlue assay used for the culture of primary CLL cells in collagen scaffolds, as such measurements were biased–it was not possible to determine whether the observed drops in the metabolic levels were caused by a lower number of cells present in the scaffold or by their lower metabolic activity.

BMSC lines grown in collagen scaffolds showed physiological phenotype, i.e., similarly to the conventional culture (Figs. 3A, 3B), they were spindle-shaped or polygon-shaped and adhered to the material (Figs. 3E, 3F). BMSCs were able to grow on fibers and walls of the scaffold septa, forming synapses in multiple directions (Figs. 3E, 3F) and not only in one plane, as seen in the conventional culture of adherent cells (Figs. 3A, 3B). B cell lines had similar morphology both in conventional (Figs. 3D, 3C) as well as 3D scaffold-based culture (Figs. 3G, 3H)–they were of a rounded shape and partially formed clumps. Contrary to the conventional culture, the suspension cell lines could interact with the surface in all dimensions.

The main advantage of the 3D scaffold culture was seen in micrographs of co-cultured M2-10B4 and primary CLL cells (Figs. 4A–4E). Similar to monoculture, BMSCs had physiological adherent phenotype and formed protrusions, lamellipodia or filopodia. However, the interaction of cells occurred not only between the top and the bottom of the cells. The suspension-growing cells were interacting with the surface and with M2-B104 cells at least in the depth up to 200 μm, as seen in 3D projection (Fig. 4B). Moreover, primary CLL cells were mostly seen in the proximity of M2-10B4 cells and only rarely solitarily. Presumably, there is a need for interactions of these cell types, which are additionally supported by the 3D platform in all dimensions.

Figure 4 Confocal micrographs of primary CLL cells co-cultured with M2-10B4 in collagen scaffolds.

CLL cells–blue, M2-10B4–green, (A) maximum intensity projection of all layers, (B) Three-dimensional view of the same scaffold, (C) one layer of the same scaffold, transmission channel added, (D, E) detail of the cells growing in scaffolds, (D) maximum intensity projection of several layers, (E) only one layer of (D), with transmission channel added.

Culture in CMC-PEG gel

As this particular mixture was not previously used for cell culture, optimizations needed to be made. The ideal dilution of the stock CMC-PEG gel was determined by culturing cells of six CLL patients (Table S1) in four different CMC-PEG concentrations for 3 days. The fresh-frozen cells had different levels of viability after thawing, ranging from 83.9% to 98.1% (Table S1). Each day, the gel-cultured cells were first examined by transmitted-light microscopy (Fig. S2); then, the AlamarBlue solution was added to measure metabolic activity (Fig. 5).

Figure 5 Difference in metabolic rates of primary CLL cells grown in four dilutions (60×, 24×, 12×, 8×) of CMC-PEG gel in comparison with conventional culture (i.e., no gel).

Measured by AlamarBlue, ΔRFU, difference in relative fluorescence units (RFU of cells cultured in gel diminished by the average RFU of cells cultured in medium with no gel); i.v., initial viability of cells. N = 3 biological replicates.

The micrographs showed that primary CLL cells cultured in gel formed clumps (Fig. S2). In no gel, 60×, 24×, and 12× diluted gel, the cells were mostly observed in one layer at the bottom of the flask. When the gel was diluted 8×, the cells were distributed in several layers, which also resulted in cells being more distant from each other compared to other dilutions. However, in 8× diluted gel, the cells could interact in more directions, contrary to lower CMC-PEG concentrations, more resembling the 2D culture. No dilution caused any visible cellular damage or changes in the primary CLL cells’ shape, i.e., they were still round, similar to culture without gel.

As for the AlamarBlue results (Fig. 5), cells with the >93% initial viability appeared to benefit from the CMC-PEG gel at any dilution (positive ΔRFU values), while cells with lower viability responded variably depending on the amount of gel. CMC-PEG at 24× dilution supported the highest metabolic activity of CLL cells in four out of six patients, especially on the first and third day. In the same four patients, 8× diluted CMC-PEG each day resulted in the lowest metabolic activity compared to other dilutions. The situation was the opposite for Pt03 cells, which were initially the most viable (98.1%) of the six samples: 8× diluted CMC-PEG caused the highest metabolic activity, while 24× diluted gel resulted in the lowest values. Therefore, 8× and 24× dilutions were considered for further optimizations.

The optimal dilution of CMC-PEG gel was further tested in a setting more resembling a CLL microenvironment: primary CLL cells were co-cultured with M2-10B4 cells in a medium supplemented with IL-4 and CD40L (as substitutes of T cell interactions). The co-culture could not be studied by AlamarBlue, as it would not be possible to determine the contribution of each cell type to the overall result. It was therefore examined by confocal microscopy (Fig. 6). First, the cells were imaged without propidium iodide (Figs. 6A–6C), as it is cytotoxic and could affect cell fitness after a long-term cell exposure (Chiaraviglio & Kirby, 2014). Then, propidium iodide was added to determine the cell viability (Figs. 6D–6F, Fig. S3).

Figure 6 Confocal micrographs of primary CLL cells co-cultured with M2-10B4.

CLL cells–blue, M2-10B4–green; cells were cultured (A, D) conventionally, in (B, E) 24× or (C, F) 8× diluted CMC-PEG gel. (A–C) A detailed view, populations stained only with Cell Trace; (D–F) More distant view, populations stained with Cell Trace and dead cells stained by propidum iodide (red).

M2-10B4 cells had an adherent phenotype in the medium with no gel (Figs. 6A, 6D) and in the medium with 24× diluted gel (Figs. 6B, 6E). In 8× diluted gel, the cells were mainly of a round shape, not attached to the bottom of the culture dish (Figs. 6C, 6F). Similar to collagen scaffolds, primary CLL cells accumulated close to M2-10B4 cells, which was most noticeable in 24× and 8× diluted gel (Figs. 6B, 6F). The viability of CLL cells was the highest in conventional culture and 24× diluted gel (Fig. S3). Thus, the 24× dilution (i.e., 2.50 g/L (w/v) CMC and 2.08 g/L (w/v) PEG) was selected for further experiments.

In addition, we used this dilution for monocultures of cell lines and observed their morphology with transmitted-light microscopy (Figs. 3I–3L). Adherent cell lines HS-5 and M2-10B4 partially adhered to the surface (Figs. 3I, 3J), although less than in medium with no gel (Figs. 3A, 3B). Suspension cell lines MEC-1 and HG-3 formed bigger clumps in CMC-PEG gel (Figs. 3K, 3L) compared to the conventional culture (Figs. 3C, 3D). Taken together, we assume that CMC-PEG gel promotes cell-to-cell contacts in suspension cell lines by the formation of cell clusters, but at the same time, the presence of gel could lead to decreased adhesion of anchorage-dependent cells.

Comparison of 2D, gel-based, and scaffold-based cell culture

To evaluate whether primary CLL cells prosper the most in conventional culture, collagen scaffolds, or the presence of CMC-PEG gel, we performed another set of measurements with cells from 15 CLL patients. The cohort cases were divided into five groups of three based on their mutational status of rearranged IGHV and the TP53 and NOTCH1 genes (Table S1). CLL cells either grew in monoculture supplemented with IL-4 and CD40L or in co-culture with M2-10B4 cells and IL-4 and CD40L. CLL monoculture, grown either conventionally or in CMC-PEG gel, was studied by AlamarBlue on days 1, 2, 4 and 7 and transmitted-light microscopy on day 7 (prior to the AlamarBlue assay). Primary CLL cells co-cultured with M2-10B4 cells conventionally, in CMC-PEG gel, or collagen scaffolds were subjected to the RT-qPCR analysis on day 2 to determine the expression level of genes involved in apoptosis, adhesion, and cell-cell interactions.

Primary CLL cells responded to the culture in CMC-PEG gel with variable changes in metabolic activity (Fig. 7, for corresponding absolute values, see Fig. S4). The RFU values of cells in CMC-PEG gels ranged from 61.86% to 169.57% of the RFU in matched conventional cultures, with a median of 115.20%; meaning there was mostly a positive effect of CMC-PEG on cell metabolism. When the median from all comparisons was calculated for each patient (i.e., from each day, and each biological replicate; Table 2), a negative response to CMC-PEG was seen only in one patient (Pt16). In 13 patients, the CMC-PEG positively influenced the metabolism (i.e., the median of RFU comparisons was >100%). When the samples were grouped based on their distinct genetic features, the elevation of metabolic activity in CMC-PEG gels was seen in each group, most notably in samples with (un)mutated IGHV, and wild type TP53 and NOTCH1.

Figure 7 Comparison of the metabolic levels of 15 CLL patient samples harboring different genetic features (% of RFU measured in conventional culture).

Measured by AlamarBlue, CMC-PEG gel stock dilution 24× was used. Unmut, unmutated; mut, mutated; wt, wild type; RFU, Relative Fluorescence Units; i.v., initial viability. N = 3 biological replicates for each culture.

Table 2 Min, max, and median values of metabolic activity of CLL cells cultured in 24× diluted CMC-PEG compared to CLL cells cultured conventionally.

Group	IGHV	TP53	NOTCH1	Patient ID	Min (%)	Max (%)	Median (%)	Median of a group (%)	Median of all (%)	
Gr1	Unmut	Mut	Mut	Pt08	87.52 (D7)	160.04 (D2)	119.22	115.59	115.20	
Pt09	87.20 (D7)	129.69 (D1)	113.53	
Pt10	84.16 (D7)	142.67 (D1)	119.45	
Gr2	Unmut	Mut	WT	Pt11	94.45 (D4)	154.08 (D7)	119.37	112.63	
Pt12	66.57 (D7)	121.40 (D4)	109.87	
Pt13	71.20 (D7)	150.57 (D2)	111.08	
Gr3	Unmut	WT	Mut	Pt14	91.44 (D2)	156.10 (D4)	116.57	111.44	
Pt15	83.33 (D7)	165.61 (D1)	114.91	
Pt16	87.41 (D4)	149.85 (D4)	98.47	
Gr4	Unmut	WT	WT	Pt17	73.02 (D7)	169.57 (D4)	137.27	126.54	
Pt18	ND	ND	ND	
Pt19	61.86 (D7)	150.59 (D4)	113.55	
Gr5	Mut	WT	WT	Pt20	111.75 (D2)	149.02 (D7)	121.35	120.48	
Pt21	77.37 (D7)	157.81 (D2)	108.48	
Pt22	111.49 (D4)	143.47 (D7)	122.18	
Note:

Unmut, unmutated; Mut, mutated; WT, wild type; D, day, ND, not determined (as the absolute RFU values were nearly zero in both materials). The calculations can be seen in the accompanying raw data files (https://doi.org/10.5281/zenodo.13933534). Each measurement (RFU value) was compared with the mean of RFUs of biological replicates (n = 3) of conventional culture measured in the same patient and same day. This resulted in three percentual comparisons with conventional culture for each day. The median for each patient is a median of all these comparisons for each biological replicate (n = 3) of all four days (i.e., median of 3 × 4 = 12 comparisons in total). Median of a group of three patients is a median of 3 × 3 × 4 = 36 comparisons in total. Median of all (i.e., all 15 patients and their three comparisons for each of 4 days) is a median of 15 × 3 × 4 = 180 comparisons in total.

It is worth mentioning that the nature of the gel-induced metabolic changes evolved in several patients–while at one time point, the CMC-PEG was associated with an increase, on another day, it was linked to a decrease in cell metabolism. Specifically, in eight out of 15 patients (Pt08, 09, 12, 13, 16, 17, 19, 21), medians of comparisons between RFU of cells cultured conventionally and in CMC-PEG gels were >100% at 2–3 time points and <100% at 1–2 time points. Moreover, in six out of these eight patients, a negative effect was observed on day 7, suggesting that the initial benefit of CMC-PEG might develop into a disadvantage for some cells after a long-term (i.e., a minimum of 7 days) exposure. In the remaining patients (Pt 10, 11, 14, 15, 20, 22), the CMC-PEG was associated with increased metabolic activity at each time point (i.e., a median of the % RFU in matched conventional cultures at each day was >100%).

As seen in our initial experiments (Figs. 3K, 3L, Fig. S2, Fig. 6), primary CLL cells formed clumps in CMC-PEG gel. A similar effect was also seen in CLL cells of the additional 15 patients cultured in CMC-PEG (Fig. 8). Surprisingly, cells of patient Pt12 formed more clumps in the medium without gel than in CMC-PEG–that could also explain the higher metabolic activity in conventional culture rather than in the gel on day 7 (median: 70.93% of the RFU in conventional culture). In addition, when the clumps promoted by CMC-PEG were too large, as seen in Pt09, it was also accompanied by lower metabolic activity compared to the gel-free medium. This suggests that the formation of cell aggregates rather than the presence of the gel might be associated with metabolic activity. We did not see any common effect of the CMC-PEG gel on cell morphology or formation of clumps based on the IGHV, TP53, and NOTCH1 statuses.

Figure 8 Micrographs of cells from the 15 CLL patients, harboring different genetic features, on day 7 of their culture.

CMC-PEG gel stock dilution 24× was used. Unmut, unmutated; mut, mutated; wt, wild type. Scale bar denotes 200 μm. N = 3 biological replicates for each culture.

Next, we measured the expression of MYC, VCAM1, MCL1, CXCR4, and CCL4 genes in primary CLL cells co-cultured with M2-10B4 cells, IL-4 and CD40L. GUSB and HPRT1 were used as endogenous controls. None of the designed primers had targets in murine RNA or unseeded scaffolds, as shown in Fig. S3. We compared whether there was any difference in the expression of these genes in primary CLL cells cultured 3D vs. 2D and whether the mutational statuses of IGHV, TP53, or NOTCH1 had any impact.

Our findings suggest that the introduction of the materials into the culture can significantly alter the expression of genes in primary CLL cells (Fig. 9, Fig. S5). It is worth noting that Group 4 (IGHV unmutated, TP53, and NOTCH1 wild type) was analyzed in a separate experiment; therefore, its outlying values in gene expression of MYC and VCAM1 genes could have been batch-related. If these outliers were omitted, compared to conventional culture, (i) expression of VCAM1 was reduced in CLL cells both in CMC-PEG and collagen scaffolds, (ii) MCL1 expression increased in CLL cells in CMC-PEG but decreased in collagen scaffolds, (iii) CXCR4 was downregulated in collagen-cultured and upregulated in CMC-PEG cultured CLL cells, and (iv) CCL4 expression was higher in CLL cells growing in collagen scaffolds (for significance, see Table 3).

Figure 9 Expression levels of five genes studied in five groups of three patients.

15 patients in total, all grouped in upper row. Their cells were cultured conventionally (control), in collagen scaffolds or 24× diluted CMC-PEG gel. Unmut, unmutated; mut, mutated; wt, wild type. N = 3 biological replicates for each culture.

Table 3 Statistical analysis of qPCR results: P-values. Statistically significant values (P < 0.05) are highlighted in bold.

Group	Genetic features	MYC	VCAM1	MCL1	CXCR4	CCL4	
IGHV	TP53	NOTCH1	CMC-PEG gel vs. Conventional culture	
All	Both	Both	Both	0.0115 (incl. Gr4)
0.1887 (excl. Gr4)	0.6788 (incl. Gr4)
6.57E–11 (excl. Gr4)	0.0053	3.97E–05	0.1875	
Gr1	Unmut	WT	Mut	0.2437	3.38E–05	0.1843	0.1119	0.1093	
Gr2	Unmut	Mut	Unmut	0.0025	6.68E–06	0.0059	0.0032	0.8153	
Gr3	Unmut	WT	Mut	0.7315	0.0490	0.8690	0.1361	0.7431	
Gr4	Mut	WT	WT	2.37E–06	6.04E–06	0.7698	0.6596	0.2011	
Gr5	Unmut	WT	WT	0.0293	0.0019	0.0441	9.39E–06	0.2758	
Group	IGHV	TP53	NOTCH1	Collagen scaffolds vs. Conventional culture	
All	Both	Both	Both	0.7629	0.0043	0.0060	0.0136	0.0003	
Gr1	Unmut	WT	Mut	0.9451	6.31E–05	0.0293	0.2287	0.5678	
Gr2	Unmut	Mut	Unmut	0.0057	0.0016	0.0877	0.4964	0.0032	
Gr3	Unmut	WT	Mut	0.2759	0.0924	0.0019	0.0010	0.9811	
Gr4	Mut	WT	Wt	0.4025	0.1548	0.3524	0.9599	0.0493	
Gr5	Unmut	WT	Wt	1.66E–06	0.0122	0.1402	0.1891	0.0496	
Note:

Unmut, unmutated; Mut, mutated; WT, wild type; Gr, group; incl., including; excl., excluding. N = 3 biological replicates for each culture.

If significant changes in gene expression were observed in separate groups of patients, they matched with the change seen in all patients grouped (Fig. 9, Table 3). The only exception was MYC expression: compared to conventional culture, it was lower in patients of Group 2 (IGHV unmutated, TP53 mutated, NOTCH1 wild type) and higher in patients of Group 5 (IGHV mutated, TP53 wild type, NOTCH1 wild type), each time in both 3D culture types.

To further address the response of primary CLL cells to different culture conditions, we evaluated the viability of CLL cells acquired from five additional patients (Fig. 10). The primary CLL cells were co-cultured with M2-10B4, IL-4, and CD40L in three environments: conventional 2D culture, collagen scaffold, and CMC-PEG gel. Considering the median viability, CLL cells cultured in collagen scaffolds exhibited higher viability than in conventional culture for three out of the five tested patients, while the CMC-PEG gel resulted in higher viability for four out of five patients. In conventional 2D culture, the median viability across the patients was 87.46%, with individual patient values ranging from 75.17% to 92.93%. In the CMC-PEG gel culture, a slightly higher median viability of 90.96% was observed, with individual patient cells’ viability ranging from 81.80% to 92.47%. Collagen scaffolds demonstrated a comparable median viability of 85.65%, with values ranging from 80.31% to 96.36%. The results indicate that both 3D culture systems (collagen and CMC-PEG) can support CLL cell viability similarly to the conventional 2D cultures, with the CMC-PEG gel slightly outperforming the collagen scaffold in terms of consistency across patients.

Figure 10 Viability of cells from five CLL patients on day 2 of their culture (conventional, in collagen scaffolds, or in 24× diluted CMC-PEG gel).

Counted from the micrographs of co-cultures. Raw images, cell counts, and calculations are available in the accompanying raw data files (https://doi.org/10.5281/zenodo.13933534). N = 5 biological replicates for each culture and patient; N = 25 for the grouped values from each type of culture.

Discussion

The development of physiologically relevant in vitro systems has been one of the ways to support the movement toward reducing in vivo experimentation (Clift & Doak, 2021). Although animal models cannot be replaced, it is certainly indisputable that in vitro cultures can exclude poor candidates from animal clinical trials and thus reduce burdens borne by lab animals.

In this study, we designed two static translucent 3D in vitro models of CLL–a scaffold-based model utilizing collagen and a viscous model employing CMC-PEG gel. The cytotoxicity and biocompatibility of these materials were already evaluated in previous studies: Collagen-based biomaterials are generally a frequent choice when reconstructing human tissues such as cartilage, bone, or skin (Rezvani Ghomi et al., 2021). Specifically, collagen scaffolds in this study have been previously successfully used for culturing rabbit mesenchymal stem cells (Prosecká et al., 2015; Vojtová et al., 2019) and 3T3 mouse fibroblasts (Babrnáková et al., 2019). CMC served in culture of hepatocellular carcinoma cells (Badekila, Rai & Kini, 2022), hematopoietic stem and/or progenitor cells (Agis et al., 2010; Tavakol et al., 2021), bone-marrow-derived stromal cells (Clarke et al., 2007; Tavakol et al., 2021), primary osteoblasts, fibroblasts and endothelial cells (Metzger et al., 2011), osteosarcoma cells (Priya et al., 2021), or porcine aorta smooth muscle cells (Lee et al., 2015). Last but not least, materials containing PEG were engineered for the culture of T cells (Pérez del Río et al., 2020; Santos et al., 2022). All these studies demonstrated the biocompatibility of the materials, which was also seen in our study. We observed that collagen scaffolds were capable of long-term culture (>60 days) of cell lines relevant to CLL studies. The collagen material support also increased the viability of primary CLL cells in three out of five patients. Moreover, in 13 of the 15 patients, there was a positive influence of the CMC-PEG gel on the metabolic levels of CLL cells, and in four of five additional patients, the CMC-PEG positively influenced the primary CLL cell viability. However, one has to be cautious when selecting a working concentration of CMC-PEG gel. When culturing CLL cells, we observed that higher concentrations of CMC (i.e., 7.5 g/L, w/v) negatively affected the viability and adherence of cells, which was not seen at 2.5 g/L (w/v) of CMC. Previous studies (Sen, Kallos & Behie, 2002; Clarke et al., 2007; Agis et al., 2010) have also shown that in high concentrations (~6.0 g/L and more), CMC can lead to proliferation inhibition or cytotoxicity. However, lowering gel concentration leads to the CLL culture being more identical to the 2D approaches, considering the spatial organization of cells. Thus, one has to find an ideal balance between material stiffness, viscosity, and cytotoxicity.

PEG and CMC aid in the formation of cell spheroids by increasing the viscosity of the culture medium (Sen, Kallos & Behie, 2002; Ariyoshi et al., 2020; Velasco-Mallorquí, Rodríguez-Comas & Ramón-Azcón, 2021). Scaffold-based culture can also support the formation of cell aggregates (Unnikrishnan, Thomas & Ram Kumar, 2021; Cao et al., 2022). Microscopically, we saw cell clumps in both types of tested 3D culture. The expression of adhesion molecule VCAM1 was, however, lower in CLL cells in both CMC-PEG and collagen scaffolds compared to conventional culture, in which the cell aggregates were not as prominent. Previous research suggested that VCAM-1 might play a role in enhanced adhesion of the malignant cells to other cells or tissues (Reuss-Borst et al., 1995) and showed that cells forming aggregates have higher VCAM1 gene expression (Ran et al., 2022). However, the generation of the aggregates in the published study was mediated by the interaction of surface molecules (Ran et al., 2022). In our materials, the formation of CLL cells’ aggregates might have been mechanistically driven, indicating that CLL cells no longer required the expression of adhesion molecules, as the materials themselves facilitated the intercellular adhesion.

Compared to 2D culture, we anticipated modulated expression of both CXCR4 and CCL4 in primary CLL cells co-cultured with M2-10B4 in a 3D environment, as these molecules are associated with the interaction between BMSCs and CLL cells (Burger, Burger & Kipps, 1999; Trimarco et al., 2015). The expression of CXCR4 in CMC-PEG gel and CCL4 in collagen scaffolds was indeed found to be upregulated. Conversely, CXCR4 was downregulated in collagen scaffolds, and gel-cultured cells did not show any alteration in CCL4 expression compared to conventional culture. Low expression levels of CXCR4 were previously (Burger, Burger & Kipps, 1999; Okkenhaug & Burger, 2016) seen in the proliferating CLL cells in bone marrow and lymph nodes, while the circulating CLL cells were found to express high levels of CXCR4. Another study (Palma et al., 2018) stated that CLL4 was more highly expressed in lymph node CLL cells as compared to peripheral blood. This suggests that the signals delivered in the collagen scaffolds resemble the situation seen in lymphoid organs, while the co-culture in the CMC-PEG gel rather corresponds to interactions in peripheral blood. Replication of in vivo stimuli is crucial for potential applications of these 3D CLL models (e.g., drug testing) as, for example, CXCR4-mediated interactions hold cancer cells within a protective tumor microenvironment and are linked to resistance to therapeutic agents (Philipp-Abbrederis et al., 2015).

Additionally, primary CLL cells co-cultured with BMSCs in collagen had lower expression of the MCL1 gene than in conventional culture. Mcl-1 is critical for CLL cell viability in MSC-co-cultures (Kurtova et al., 2009). More specifically, the resulting protein of MCL1 transcript variant 2, which we studied, promotes apoptosis (Bae et al., 2000). Decreased apoptosis in collagen scaffolds could be explained by the presence of collagen receptors on the surface of circulating CLL cells, namely discoidin domain receptor (DDR1), which may act as a sensor for stromal collagen and provide a supportive stimulus (Barisione et al., 2017). On the other hand, we observed increased levels of MCL1 gene expression in cells cultured in CMC-PEG than in cells grown without gel. The pro-apoptotic phenotype of CLL cells in CMC-PEG could be balanced by lower expression of other pro-apoptotic genes or higher expression of anti-apoptotic genes since the metabolic activities measured on day 2 (when the samples were collected for RNA isolation) did not suggest that presence of 24× diluted CMC-PEG gel harmed the CLL cells. However, in several samples, the upregulation of MCL1 could be an early sign of induced apoptosis, which became evident on day 7 as a measurable reduction in metabolic activity of cells cultured in the gel compared to conventional culture. It is also possible that the formation of cell aggregates acted in two ways–while it might have advantageously promoted pro-survival signaling, it also facilitated the spread of death signals from apoptotic to healthy cells, e.g., via gap junctions (Decrock et al., 2009).

Furthermore, a different expression of MYC in a 3D culture compared to a conventional one was observed in patients grouped according to their genetic features. In 3D culture, MYC expression was downregulated in patients of Group 2 (IGHV unmutated, TP53 mutated, NOTCH1 wild type) and upregulated in patients of Group 5 (IGHV mutated, TP53 wild type, NOTCH1 wild type). We expected NOTCH1 mutational status would alter MYC gene expression, as its activation might confer cell growth and/or proliferation advantages to CLL cells (Pozzo et al., 2017). Moreover, MYC is a transcriptional target of the NOTCH1 activation complex in CLL, and modulation of NOTCH1 signaling influences MYC transcript levels (Palomero et al., 2006; Fabbri et al., 2017). However, diverse alterations in MYC expression were observed in patients with wild-type NOTCH1. Hence, mutations in TP53 or IGHV could have played a role. Additional comparative analysis of our findings with previously published results (Sbrana et al., 2021), showing downregulated expression of MYC in 3D-cultured cells, would be required to establish whether this is a reoccurring pattern.

Summary/conclusions

Our study aimed to compare different methods to culture BMSC and CLL cell lines and genetically characterized CLL primary cells, the latter either mono-cultured or co-cultured with murine BMSC cell line M2-10B4. We have developed two 3D in vitro culture platforms: collagen scaffolds and CMC-PEG gel. We optimized the methodology and studied whether the novel approaches in CLL culture could enhance cell-to-cell and cell-to-matrix interactions, cell viability, metabolic activity, and modulate gene expression related to apoptosis and intercellular interactions.

Each material has its advantages in terms of practicality. Collagen scaffolds are biodegradable and easy to handle, wash and transfer. However, their preparation is time-consuming, and they lack the ability to accommodate a precise number of cells. On the other hand, CMC-PEG gel is easy and fast to prepare and allows seeding with a well-known number of cells. Nonetheless, working with this material can be challenging due to its viscosity. Both materials are biocompatible and versatile, enabling various applications such as microscopy, microtiter plate assays, or RNA expression analyses. Culturing in both these models leads to the formation of cell clumps and can increase metabolic levels of CLL cells (in CMC-PEG gel) or lower the expression of the pro-apoptotic genes (in collagen scaffolds). The materials also promote the BMSC-CLL interactions in the co-culture models, which can increase the CLL cells’ viability.

In summary, this study demonstrates the potential of 3D culture platforms, such as collagen scaffolds and CMC-PEG gel, to enhance our understanding of BMSC-CLL interactions and provide valuable insights into CLL biology. Our methodology and findings pave the way for future investigations to elucidate mechanisms underlying the behavior of CLL cells cultured three-dimensionally.

Supplemental Information

Supplemental Information 1 Amplification plots from primer specificity tests with various templates.

The templates were following: human RNA (10 ng per reaction), murine RNA (100 ng per reaction), RNA from an unseeded collagen scaffold, RNA from an unseeded CMC-PEG gel, negative control (nuclease-free water). ΔRn – difference between normalized reporter (Rn) at the end point and at the starting point.

Supplemental Information 2 Micrographs of B cells of six CLL patients.

Cells were cultured conventionally (no gel) and in several dilutions (8×, 12×, 24×, 60×) of CMC-PEG gel for 3 days. Captured with EVOS FL microscope, transmission channel. Scale bar denotes 400 μm.

Supplemental Information 3 Viability of primary CLL cells cultured conventionally, in 24× or 8× diluted CMC-PEG gel.

No gel – median 94.73% (min 91.42%, max 97.00%), 24× diluted gel – median 95.29% (min 86.43%, max 96.84%), 8× diluted gel – median 89.92 % (min 83.48%, max 96.78 %). N=6 biological replicates. The calculations can be seen in the accompanying raw data files (https://doi.org/10.5281/zenodo.13933534). <!--[if !supportAnnotations]--> <!--[endif]-->

Supplemental Information 4 Metabolic levels of the 15 CLL patient samples harboring different genetic features – absolute values.

Measured by AlamarBlue, CMC-PEG gel stock dilution 24× was used. Unmut – unmutated, mut – mutated, wt – wild type, RFU – Relative Fluorescence Units, i.v. – initial viability. N=3 biological replicates for each culture.

Supplemental Information 5 Expression levels of five genes studied in 15 patients, whose cells were cultured conventionally (control), in collagen scaffolds or 24× diluted CMC-PEG gel.

N=3 biological replicates for each culture.

Supplemental Information 6 MIQE checklist, RT-qPCR details.

Supplemental Information 7 Characteristics of patients and their CLL cells used in experiments.

M – male, F – female, Mut – mutated, Unmut – unmutated, WT – wild type, NA – not available.

Supplemental Information 8 Media used for culture of cells (Biosera, Nuaille, France).

Supplemental Information 9 Series of ImageJ commands used for counting viability of one cell type in a co-culture of two cell types.

Supplemental Information 10 Validated human-specific primers used for qPCR with no targets in Mus musculus and Bos taurus.

We would like to express our gratitude to the team of Vitezslav Bryja (namely Pavlina Janovska, and Milena Marakova; from the Institute of Experimental Biology, Faculty of Science, Masaryk University, Brno, Czech Republic) for providing MYC and VCAM1 primers. We also thank the research team of Richard Rosenquist Brandell (Karolinska Institutet, Stockholm, and previously, the Department of Immunology, Genetics and Pathology, Uppsala University, Sweden) for kindly providing us with the HG-3 cell line.

Additional Information and Declarations

Competing Interests

Author Contributions

Human Ethics

Data Availability

The authors declare that they have no competing interests.

Hana Svozilova conceived and designed the experiments, performed the experiments, analyzed the data, prepared figures and/or tables, authored or reviewed drafts of the article, prepared the biomaterials, and approved the final draft.

Lucy Vojtova conceived and designed the experiments, prepared figures and/or tables, authored or reviewed drafts of the article, and approved the final draft.

Jana Matulova conceived and designed the experiments, performed the experiments, authored or reviewed drafts of the article, prepared the biomaterials, and approved the final draft.

Jana Bruknerova conceived and designed the experiments, performed the experiments, authored or reviewed drafts of the article, and approved the final draft.

Veronika Polakova conceived and designed the experiments, performed the experiments, authored or reviewed drafts of the article, prepared the biomaterials, and approved the final draft.

Lenka Radova analyzed the data, prepared figures and/or tables, authored or reviewed drafts of the article, and approved the final draft.

Michael Doubek conceived and designed the experiments, authored or reviewed drafts of the article, and approved the final draft.

Karla Plevova conceived and designed the experiments, authored or reviewed drafts of the article, and approved the final draft.

Sarka Pospisilova conceived and designed the experiments, authored or reviewed drafts of the article, and approved the final draft.

The following information was supplied relating to ethical approvals (i.e., approving body and any reference numbers):

The project was approved by the hospital’s ethical committee (date of approval: 4th April 2018, registration number Sup 8/18, University Hospital Brno).

The following information was supplied regarding data availability:

The data is available at Zenodo: Hana Svozilová, Lucy Vojtová, Jana Matulová, Jana Bruknerová, Veronika Poláková, Lenka Radová, Michael Doubek, Karla Plevová, & Šárka Pospíšilová. (2024). Raw data for the article "In vitro culture of leukemic cells in collagen scaffolds and carboxymethylcellulose-polyethylene glycol gel" [Data set]. Zenodo. https://doi.org/10.5281/zenodo.13933534.

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
