# Peer review of "In vitro culture of leukemic cells in collagen scaffolds and carboxymethyl cellulose-polyethylene glycol gel"

_PeerJ, doi:10.7717/peerj.18637_

## Round 0.1 · original submission · Major Revisions

Dear authors,

The manuscript is overall well written and addresses an important topic. However, I encourage you to carefully consider all the criticisms and comments raised by the reviewers and to provide appropriate responses.

·

Basic reporting

The manuscript titled “In vitro Culture of Leukemic Cells in Collagen Scaffolds and Carboxymethyl Cellulose-Polyethylene Glycol Gel” aims to enhance the in vitro culture of primary chronic lymphocytic leukemia (CLL) cells, which tend to die rapidly when grown outside their natural environment. In this study, collagen scaffolds and a mixed gel of carboxymethyl cellulose (CMC) and polyethylene glycol (PEG) were used to culture CLL cells. The study specifically aims to evaluate cell viability, metabolic levels, morphology, and changes in gene expression in 3D culture compared to conventional 2D culture.
Both materials promoted cell-to-cell and cell-to-matrix interactions due to the scaffold structure and the generation of cell aggregates. The expression of the adhesion molecule VCAM1 was lower in 3D cultures than in conventional cultures. Additionally, CXCR4 was upregulated in CMC-PEG gel, and CCL4 was upregulated in collagen scaffolds. Cells cultured with MSCs in collagen showed lower expression of the MCL1 gene compared to conventional culture.

The manuscript is well-written, and the topic is significant.

Experimental design

Here are some suggestions to improve the clarity of the text and strengthen the results:

• In line 343, the authors state that fresh frozen cells derived from patients had different levels of viability after thawing, ranging from 83.9% to 98.1%. Could the authors present this data?
• The authors co-cultured CLL cells with M2-10B4 (line 363). Could the authors specify which CLL cell line they chose for the experiments?
• The authors could not use the Alamar Blue assay to study the cell viability of the co-cultures because it is not possible to determine the contribution of each cell type, so they added propidium iodide (Fig 6). After how much time was it added?
• To determine the cellular viability of the co-cultured cells, the authors could perform a time-based experiment and analyze by cytometry how the fluorescence intensity of both co-cultured cell lines varies over time compared to the control where the cell lines are cultured individually.
• In Table 2, could the authors better explain how they obtained the median?
• In Figure 9, could the authors improve the quality of the image by enlarging it? Additionally, as a suggest, the authors could add significance using asterisks (i.e., *p < 0.05, **p < 0.005, ***p < 0.0005) to make it easier to understand. Furthermore, since group 4 (IGHV unmutated, TP53, and NOTCH1 wild type) was analyzed in a separate experiment, the authors could consider including this data in the supplementary materials.

Validity of the findings

The research brings innovation and rigor to the field of chronic lymphocytic leukemia. The reported data are valid and pave the way for new in vitro research. The conclusions align well with the research objectives and results. Therefore, apart from the minor suggested modifications, I consider the research both valid and innovative.

Additional comments

1. In line 369 of the text, the authors refer to Figure S3, not S2.
2. In line 371 of the text, the authors refer to Figures 6B and E, not F.
3. In line 84 of the text, add "in" before "a three dimensional".
4. In line 366 of the text, delete "in" after "projection".

Reviewer 2 ·

Basic reporting

In the present manuscript Svozilov et al. investigated the behavior of CLL cells and bone marrow stromal cells in two types of material: (i) solid porous collagen scaffolds and (ii) gel composed of carboxymethyl cellulose and polyethylene glycol (CMC-PEG). Authors characterized the materials and demonstrated that both promoted cell-to-cell and cell-to- matrix interactions and support higher metabolic activity of CLL than in conventional culture. Interestingly for the study they used an heterogenous cohort of patients with CLL divided into five groups of three based on their mutational status of rearranged IGHV and the TP53 and NOTCH1 genes.
The paper is in general clear and well written however there are some criticism at methodological level that need to be addressed to support the conclusions.

Experimental design

Major comments:
1. Scaffold characterization: the two materials were analyzed in different ways. The collagen scaffold (only) was tested with cell lines not used in the following experiments with Alamar Blue, why? Primary CLL cells viability was not tested in the collagen scaffold (line 316 to 322) due to bias in Alamar measurements, while in CMC-PEG gel the viability was evaluated qualitatively with PI in co-cultures with BM stromal cell lines (lines 362 to369). It would be better to evaluate viability with the same method to compare the materials. How is it calculated the viability in Fig. S3? The scaffolds are not acquired in their entirety (line 335), how are the cells in the middle of the scaffold? From fig S2 how can be assumed that in some concentrations of the gel cells lay only in one or several layers? (line 346 to 352) please explain. In general, it would be better to uniform the analysis of the two scaffolds regarding cell viability.
2. Comparison of 2D, gel based, and scaffold-based cell culture: the paragraph introduces a comparison in cell proliferation of CLL cells cultured in 2D, collagens scaffold and gel scaffold. But in the results showing viability and morphology the comparison is only between 2D and gel scaffold (fig. 7, fig. S4). Please explain if there is a reason for this choice.
3. Gene expression analysis: It would be better to move this part to a different paragraph from the comparison of morphology and viability. PCR results are better visualized in supplement figure S5 than in Fig.9. In Fig. S5 CLL patients are divided per mutational status with statistical significance this should be in the main images rather than in supplementary, in this way it would be easier to follow the logical sequence of the results. At the end authors considered the results of the different patients' groups altogether. If the results are shown also differentiating the mutational status of the patients it should be also reported an explanation on statistical differences or trend in the gene expression profile analyzed between different groups. It would be interesting to increase the number of patients per group of mutational status to draw conclusions also about gene expression differences between them.

Validity of the findings

In summary this study demonstrates the potential of 3D culture platforms, such as collagen to enhance our understanding of BMSC-CLL interactions and provide valuable insights into CLL biology paving the way for future investigations to elucidate mechanisms underlying the behavior of CLL cells cultured in 3D.

---

## Round 0.2 · accepted · Accept

The authors have adequately addressed the comments made by the reviewer

·

Basic reporting

The authors have answered all the requests so for me the article can be accepted

Experimental design

no comment

Validity of the findings

no comment